# Proton Beam Therapy for Esophageal Cancer

**DOI:** 10.3390/cancers14164045

**Published:** 2022-08-22

**Authors:** Jea Giezl N. Solidum, Raniv D. Rojo, Jennifer Y. Wo, Edward Christopher Dee

**Affiliations:** 1College of Medicine, University of the Philippines Manila, Manila 1000, Metro Manila, Philippines; 2Department of Radiation Oncology, Massachusetts General Hospital, 100 Blossom St., Boston, MA 02114, USA; 3Harvard Medical School, Boston, MA 02115, USA; 4Department of Radiation Oncology, Memorial Sloan Kettering Cancer Center, New York, NY 10065, USA

**Keywords:** proton therapy, radiotherapy, esophageal cancer, radiation exposure

## Abstract

**Simple Summary:**

Early-stage esophageal cancer is managed surgically, with the addition of radiotherapy for locally advanced disease. Current photon-based radiotherapy results in a high treatment-related complications, due to proximal organ involvement. The anatomic location of the esophagus raises challenges due to the anatomical changes associated with diaphragmatic motion, weight loss, tumor changes, and set-up variability. These propelled the interest in proton beam therapy (PBT), which theoretically offers a reduction in the radiation exposure to healthy neighboring tissues with improvements in the therapeutic ratio. In this review, we present the role of PBT for esophageal cancer, including treatment planning, early clinical comparisons with photon-based techniques, ongoing trials, current challenges, toxicities, and issues of equity and health services.

**Abstract:**

Early-stage esophageal cancer is often primarily managed surgically, with the addition of radiotherapy for locally advanced disease. However, current photon-based radiotherapy regimens and surgery results in a high incidence of treatment-related cardiac and pulmonary complications due to the involvement of proximal organs at risk. In addition, the anatomic location of the esophagus raises challenges for radiotherapy due to the anatomical changes associated with diaphragmatic motion, weight loss, tumor changes, and set-up variability. These challenges propelled the interest in proton beam therapy (PBT), which theoretically offers a reduction in the radiation exposure to healthy neighboring tissues with improvements in the therapeutic ratio. Several dosimetric studies support the potential advantages of PBT for esophageal cancer treatment however, translation of these results to improved clinical outcomes remains unclear with limited clinical data, especially in large populations. Studies on the effect on quality of life are likewise lacking. Here, we review the existing and emerging role of PBT for esophageal cancer, including treatment planning, early clinical comparisons of PBT with photon-based techniques, recently concluded and ongoing clinical trials, challenges and toxicities, effects on quality of life, and global inequities in the treatment of esophageal cancer.

## 1. Introduction: Background and Current Treatment of Esophageal Cancer

In 2020, esophageal cancer cases ranked seventh among all of the other cancers, with an incidence of 604,100 cases (3.1%) and ranked sixth in mortality with 544,076 (5.5%) deaths attributed worldwide [1]. Over 90% of the esophageal cancer cases is recorded in Asia and sub-Saharan Africa [2], with men significantly affected more than women. The incidence of esophageal squamous cell carcinoma (SCC) is broadly in decline, due to improvements in diets, and the reduction in heavy alcohol use and cigarette smoking in the high-risk areas in Asia, North America, and Western Europe [3]. On the other hand, incidence rates of esophageal adenocarcinoma (AC) are rising rapidly, due to greater rates of obesity and gastroesophageal reflux and decreasing chronic H. pylori infection, inversely associated with AC [4].

The standard treatment of care for locally advanced esophageal cancer is neoadjuvant chemoradiotherapy, based on meta-analyses and the CROSS trial [5,6]. However, neoadjuvant chemotherapy appears to show decreased postoperative complications while showing a similar overall survival (OS) rate. Neoadjuvant chemoradiotherapy may also be associated with more deaths that are unrelated to disease progression, although the difference was not statistically significant [6,7,8,9]. For the patients with extensive lymph node metastasis, or non-surgical candidates, definitive chemoradiotherapy may be a reasonable option compared to trimodal surgery [1,10], specifically in SCC, with an equivalent 2-year OS rate (39.9% vs. 35.4%) and lower treatment mortality rate (12.8% vs. 3.5%; *p* = 0.03) [11]. The common regimen is a dose of 50–50.4 Gy in 5–5.5 weeks and concurrent chemotherapy of 5-fluorouracil (5FU) and cisplatin, as established by RTOG 8501 and the Intergroup study 0123 clinical trials [12,13,14]. Because locoregional recurrences tend to occur, necessitating salvage surgery [15,16], chemoradiotherapy and salvage surgery is a logical treatment scheme for the patients who prefer to avoid upfront surgery.

The current standard radiation technique used for esophageal cancer is the three-dimensional conformal radiotherapy (3DCRT). However, this is already being challenged by the use of intensity-modulated radiotherapy (IMRT) with inverse planning, which can possibly spare the adjacent normal tissues without compromising the dose targeted to the tumor, improving upon 3DCRT [17,18,19]. The esophagus is surrounded by the bilateral lungs and mediastinal organs. Radiation therapy may result in acute toxicities, such as radiation pneumonitis (RP), and late toxicities including cardiac complications, such as pericardial disease, valvular heart disease, coronary artery atherosclerosis, cardiomyopathy, arrhythmias, or deaths related to radiation exposure [6,14]. The Surveillance, Epidemiology, and End Results (SEER) program showed that there is a higher chance of cardiac death among patients after radiotherapy (RT) compared to those who did not undergo RT (*p* < 0.0001) [20]. In addition, the analysis of the results shows that 2.8%, 5.3%, and 9.4% of radiation-treated survivors of 5-, 10-, and 20- years are at an absolute risk of heart disease-related death [21]. The occurrence of cardiac complications, such as pericardial effusion (PE), depends largely on the pericardial dose. The risk of PE significantly increased with a mean pericardial dose of 26.1 Gy [22], observed asymptomatic PE with 36.2 Gy, and symptomatic PE with 43.2 Gy [23]. Hayashi et al. (2019) recently defined a cut-off volume of 280 mL of the heart irradiated with a dose higher than 50 Gy (V50), presenting a risk ratio of 16.80 (95% CI, 4.94–53.07) for cardiac toxicity [24]. The data from Memorial Sloan Kettering Cancer Center comparing standard 3DCRT and IMRT showed a significant reduction in mean pericardial dose from 28.2 to 22.9 Gy, and V30 from 61.0% to 24.8% in favor of IMRT [19]. The incidence of RP is also dependent on the radiation dose. The only study that has looked into V20, the most frequently reported dose volume histogram (DVH) parameter, suggests an increasing incidence of symptomatic RP with dose, with an observed steepening between 23–25%. Keeping V20 below 23% maintains the incidence of RP below 15% [25]. The patients who underwent IMRT, unlike 3DCRT, did not develop grade II or higher RP with a V5 greater than 71% [25], or any clinical RP with a V5 of 90.1% [26].

The recent studies have shown that, compared with IMRT, proton beam therapy (PBT) may be associated with additional decreases in the cardiac and pulmonary dose [27,28]. Therefore, PBT is an exciting area of potential improvement in the care of patients with esophageal cancer.

## 2. Rationale of PBT as Treatment for Esophageal Cancer

PBT uses charged particles which are capable of delivering a desired RT dose to the target volume, but with significantly lower entry and exit doses, reducing the radiation dose exposure of the surrounding normal tissues as compared to other photon-based techniques [29]. Photons or electron beams deposit most of their energy near the surface, thus with progressively smaller energy with depth where the tumor may be located. Beyond the tumor, the photon-based therapy continues to deposit the dose in normal tissues. PBT has a physical property modeled by the Bragg peak, which theoretically improves the sparing of the proximal and distal non-tumor tissues. The PBT-charged particles deposit a low dose near the surface, followed by a rapid peak of energy deposition just upstream of the beam’s useful range targeting the tumor, then a quick dose drop-off subsequently. This pattern results in a sharp and localized peak dose [29,30]. To target the tumor, adjustments in the initial energy of the particles and the intensity of the beam are made. The former determines how deep in the body the Bragg peak will form, while the latter determines the dose that will be deposited to the organs at risk [30]. To adjust for tumor size, techniques such as passive scatter PBT (PS-PBT) and pencil-beam scanning PBT (PBS-PBT) are used. PS-PBT was the initial PBT technology implemented and produces a “spread out Bragg-peak” (SOBP), covering the depths of the tumor by placing a variable degrader into the path of the proton beam. On the other hand, the PBS-PBT uses a more advanced technology of fast dipole magnets to transversely scan over the tumor cross-section spot-by-spot along with the fast beam energy switching to cover the tumor depths [29].

As mentioned, the evidence shows that the incidences of lung and cardiac toxicities during esophageal cancer RT treatment are radiation dose-dependent [22,23,25]. The standard external-beam RT techniques, such as 3DCRT and IMRT, deposit the RT dose in all of the tissues along the beam path, potentially resulting in RT toxicity in normal tissues in the process. With the discussed properties and advanced technologies of PBT, it may improve the therapeutic outcome for esophageal cancer, mainly through a reduction in the adverse effects [29]. Indeed, dosimetric studies comparing IMRT with PS-PBT [31,32,33] or PBS-PBT [33,34] showed the advantage of PBT over IMRT (Table 1). One retrospective study of 343 patients comparing IMRT or PS-PBT treatment delivered with two-field posterior and left posterior-oblique fields completed by Xi et al. (2017) showed mean doses to the lung and heart of 10.0 and 19.9 Gy for IMRT versus 6.5 and 11.6 Gy for PBT (*p* < 0.001), respectively. The PBT group compared to the IMRT group also had significantly lower lung V5 Gy (48% vs. 28%; *p* < 0.001), V10 Gy (32% vs. 23%; *p* < 0.001), and V20 Gy (18% vs. 11%; *p* < 0.001), as well as lower heart V30 Gy (24% vs. 19%; *p* < 0.001) [32]. Subsequently, the PBS-PBT showed a lowering of the mean pericardial dose V20 Gy, V30 Gy, and V40 Gy, as well as the mean dose received by the substructures of the heart, such as the left and right atrium, left main coronary artery, and left circumflex artery [33].

In our current era of precision medicine, PBT undeniably has a place in the precision RT for cancer treatment. PBT was first considered for ocular tumors, skull-base tumors, paraspinal tumors (chondrosarcoma and chordoma), unresectable sarcomas, and tumors that responded poorly to photon-based RT, as well as for pediatric cancer treatment [29]. With the growing literature, PBT is now starting to be included in treatment guidelines, such as in several Children’s Oncology Group (COG) protocols [40]. PBT was also supported by The American Society of Radiation Oncology since 2017 as a treatment option for solid neoplasms in children [40]. Furthermore, PBT is an option in the updated 2017 National Comprehensive Cancer Network (NCCN) guidelines for periorbital cancer, and a number of head and neck cancers (e.g., ethmoid sinus, salivary gland, maxillary sinus, nasopharynx, mucosal melanoma), when the RT dose exposure of normal tissues with photon-based therapy is deemed too high [41]. PBT in breast cancer treatment is also being considered. Given the anatomical location of the organ, the use of PBT will potentially reduce the radiation exposure of the organs in the chest and mediastinal area, particularly the lungs and heart as with esophageal cancer. However, further investigation of the short- and long-term clinical outcomes of PBT as a breast cancer treatment is needed [41]. Currently, Memorial Sloan Kettering Cancer Center is implementing a phase III randomized clinical trial of PBT vs. photon therapy for non-metastatic breast cancer, to assess whether PBT is associated with a better side-effect profile during breast cancer treatment as compared to the standard photon therapy (RADCOMP Trial; NCT02603341). Anal cancer can also potentially benefit from PBT. A recent multi-institutional prospective feasibility study showed that the rate of grade 3+ radiation dermatitis with PBT (24%) [42] and the 2-year rate of overall complete response (88%) [43] with the concurrent chemoradiation of PBS-PBT with 5FU and mitomycin C (MMC) in patients with clinical T1–4, N0–3 anal cancer favored PBT over chemotherapy alone [42]; PBT is also considered for re-irradiation in the setting of recurrent disease [44]. For example, locally recurrent anal cancers treated with PBT showed <25% experienced recurrence within the treatment field, despite demonstrating 50% positive resection margins [45], suggesting a treatment benefit associated with PBT. Further work is needed to evaluate the role of PBT in various clinical settings.

## 3. Clinical Outcomes of PBT for Esophageal Cancer

The dosimetric data support the potential advantages of PBT for esophageal cancer treatment. However, the translation of these results to improved clinical outcomes remains unclear, as the clinical data are still limited. The early studies on the use of PBT in esophageal cancer combined proton plans with photon-based RT [46,47]. Initial treatment with photon-based therapy or IMRT, especially for extended field irradiation, is chosen because esophageal SCC has a high potential for widespread lymph-node metastasis [48]. The incidence of nodal involvement is high at 42% in clinical stage T1–3N0M0 disease, such that for certain patients, it may still be necessary to irradiate a wider field, even if no lymph node metastases are radiologically detected prior to the treatment [49]. Mizumoto et al. (2010) conducted a cohort study of 51 patients with predominantly locally advanced SCC who were treated under two different regimens but who were all given PS-PBT therapy. The regimens were an initial treatment with Xray to a median dose of 48 Gy, followed by a PS-PBT boost to the tumor to a median dose of 31.7 Gy (median total dose of 76 Gy), or direct PS-PBT therapy alone to a median dose of 79 Gy. No patients had severe enough AEs that warranted a treatment break. There were also no symptomatic, late cardiopulmonary toxicities. Together with an observed five-year OS rate of 34% [47], this supports the potential efficacy of PBT.

Another study by Ishikawa et al. (2015) assessed PS-PBT with concurrent chemotherapy. The study included 40 esophageal cancer patients, the majority with SCC, treated with PS-PBT to a dose of 60 Gy in 30 fractions with concurrent 5FU and cisplatin. No grade 3+ cardio-pulmonary toxicities were observed, the two-year OS rate was 77%, and the locoregional control (LRC) was 66%, respectively [50]. A retrospective cohort study of 27 Stage III thoracic esophageal SCC by Hirano et al. (2018) similarly showed a high one-year OS and progression-free survival (PFS) after completion of PBT of 90.8 and 40.6%, respectively. A slightly higher mean pericardial dose is seen in the patients who developed grade 2 PE as compared to those who did not. However, this correlation was not statistically significant. One patient experienced grade 3 esophagitis which warranted the disruption of the PBT. Four patients (15%) who underwent PBT developed late toxicity grade 2 PE, but none developed acute or late grade 3, or worse pulmonary and cardiac toxicities [37]. The work by Takada et al. (2016) evaluated the efficacy and safety of PBT in 47 stage-III esophageal cancer patients treated with two cycles of continuous 5FU infusion (days 1–5) and a 5-h nedaplatin infusion (day 6), accompanied by X-ray and PBT; specifically, a prophylactic region was defined, based on the tumor anatomy and treated with conventional X-ray-based radiation, and a PBT boost was added to the primary tumor and involved nodes [28]. The three-year OS, PFS, and LRC rate were 59.2%, 56.3%, and 69.8%, respectively. Although two of the patients (4.3%) had esophageal stenosis, one (2.1%) had a fistula, and two (4.3%) developed RP, no one had pleural or PE [28]. All of these studies support the use of PBT for esophageal cancer (Table 2).

The reports on the clinical outcomes of PBT in a large population of esophageal cancer patients are limited. From the four PBT centers in Japan, Ono et al. (2019) recruited 202 newly diagnosed individuals with esophageal SCC and AC who received PBT, with 44.5% having an inoperable disease and 49.5% having stage III/IV cancer [48]. One patient developed grade 3 pneumonia (0.5%) and two developed grade 3 PE (1%); none developed grade 4 or greater cardiopulmonary toxicities [48]. Furthermore, three- and five-year OS rates of 66.7% and 56.3% were reported [48]. These results are higher than the previous studies on photon RT in Japan with five- and four-year OS rates of 35–37% and 24.4%, respectively [54,55,56]. The five-year local control rate was 64.4% with factors T3–4, N1–3, stages III and IV, and inoperable tumor status considered significant in the univariate analysis, with the last significant in the multivariate analysis as well [48]. Local recurrence can decrease a patient’s quality of life (QoL). Multiple studies, including meta-analysis, support that a higher RT dose leads to a higher pathological complete response [57]. Although indirectly compared, the results of a study by Suh et al. (2014) with a total dose of >60 Gy demonstrated a lower incidence of LRC and higher OS than the results shown in a study by Zhang et al. (2005), which used a total dose of >51 Gy [58,59]. Increasing the PBT dose used in inoperable esophageal cancer patients may improve the LRC.

The prospective studies directly comparing PBT and photon radiotherapy are few [48]. A retrospective cohort study of patients with stage I–III esophageal cancer was conducted at MD Anderson Cancer Center by Xi et al. (2017). This study compared PBT with IMRT. The patients were treated with radiotherapy to a median dose of 50.4 Gy in 28 fractions; the patients also received concurrent chemotherapy and no surgery [32]. PBT resulted in a favorable five-year OS rate of 41.6%, greater than the comparison arm (OS 31.6%) [32]. The OS benefit was maintained on multivariate analysis. The locoregional failure-free survival was greater (HR 0.684; *p* = 0.041) particularly in the stage III cancer patients, with no difference in AEs [32].

Most of the randomized trials of protons and photons were designed more to investigate the dose escalation safety of PBT rather than as true comparisons between PBT and IMRT. The study by Lin and colleagues, also at MD Anderson, can be considered as the first randomized trial of PBT vs. photon therapy that has achieved a positive primary endpoint (ClinicalTrials.gov identifier: NCT01512589) [60]. Lin et al. (2020) recently published the final results of their randomized phase IIB trial comparing the clinical outcomes of PBT and IMRT as the treatment for locally advanced esophageal cancer. The primary outcomes that were measured are the total toxicity burden (TTB) and PFS. TTB is a unique endpoint that takes into consideration all of the adverse events experienced by the patient over a determined span of time, weighted based on the toxicity severity [53]. The results are applicable to both surgically treated and non-surgically treated patients, potentially expanding the indications of PBT [53]. Understanding the TTB results may also help in addressing the economic aspect of PBT. AEs inadvertently lead to higher healthcare costs. The reductions in TTB will therefore improve the cost-effectiveness of PBT [32,61]. PBT is not yet included in insurance policies, as it is yet to be included as one of the official treatment options in the treatment guidelines for esophageal cancer. This places the financial burden on patients. This is reflected in the trial of Lin and colleagues. The study showed results favoring PBT over IMRT with 2.3 times lower posterior mean TTB, 7.6 times lower mean post-operative complication (POC) score, but similar three-year PFS and OS rate (51.2% vs. 50.8%; 44.5% vs. 44.5%). Despite these, the 22 PBT patients could not continue on the study owing to insurance denial [53].

Overall, the prospective and retrospective data presented support the notion that the higher the dose delivered by PBT to the tumor, coupled with the lower dose delivered to the surrounding normal organ tissues, may facilitate better OS, PFS, and LRC rates, as well as fewer acute and late toxicities and complications. In addition, the lower integral dose of PBT reduces the incidence of grade 4 lymphopenia, and is believed to be an integral factor to the higher PFS/OS rates [62,63,64]. However, the relationship between radiation-induced immunosuppression and disease-specific outcomes is yet to be ascertained [53].

Aside from the clinical outcomes, it is also important to assess the health-related QoL of patients. The cardiopulmonary late toxicities compromise the QoL of patients [37]. Several authors even described these complications as possibly life-threatening [65,66]. For the esophageal SCC patients treated with definitive CRT, one study reported 2 deaths because of acute myocardial infarction, and 8 deaths believed to be due to cardiopulmonary toxicities, among the 78 patients who achieved complete remission [65]. Considering these clinical outcomes, efforts are needed to reduce the occurrence of severe late toxicities [48], such as through the employment of PBT. However, the QoL data are very limited for the PBT treatment of esophageal cancer. A systematic review completed by Verma et al. (2018) only reported one study on head and neck cancer. The patient-reported outcomes were reportedly higher for PBT than photon-based RT for head and neck cancer [67]. Based on these limited data, PBT provides a favorable QoL/PRO profile. A more recent prospective registry series completed by Garant et al. (2019), comparing patient-reported QoL for patients with esophageal cancer treated with PBS-PBT vs. photon-based CRT, also had a similar conclusion. A preoperative CRT followed by esophagectomy was performed in 60% of the patients. A dose of 50–50.4 Gy in 25–28 fractions delivered with two posterior-oblique fields was commonly used for those who had the PBS-PBT therapy [68]. In addition to the esophageal cancer symptoms, evaluation of the patient QoL demonstrated that PBS-PBT was associated with decreased QoL deficits compared with the photon-based RT [68].

## 4. Reduction of Treatment-Related Lymphopenia with PBT

Lymphocytes are key immune response mediators against tumors [69]; the lymphocyte count is a good prognostic factor in cancer survival [70]. Subsequently, lymphopenia has been associated with worse clinical outcomes in cancer patients, including esophageal cancer [62,71,72]. The lymphocytes are extremely sensitive to radiation exposure, even at low doses [62]. Many standard fractionation courses result in a mean dose of 2 Gy delivered to the lymphocytes, which is at the higher value of its 50% lethal dose (LD50) of 1 to 2 Gy [64,73,74]. Treatment-related lymphopenia (TRL) therefore occurs in esophageal cancer patients treated with RT or concurrent CRT. A study by Zhou et al. (2019) showed that 89 out of 286 patients (31%) with stage II–IVa esophageal SCC treated with concurrent CRT developed TRL within 6 weeks from the start of therapy [75]. The TRL was seen to be associated with worse OS, disease-free survival, PFS, distant metastasis, and cancer-related deaths [31,63,72].

Several studies suggest that the use of PBT compared to photon-based techniques results in less lymphopenia (Table 3). Independent studies by Routman et al. (2019) and by Shiraishi et al. (2018) both showed that 3DCRT or IMRT were associated with a significantly higher risk of grade 4 lymphopenia (G4L) compared to PBT, with odds ratios of 5.13 and 3.45 per study, respectively [62,64]. In addition to the increased G4L with IMRT versus PBT (OR: 2.13), Fang et al. (2017) additionally showed that a greater planning target volume (OR: 3.47) also resulted in a higher incidence of lymphopenia. The radiation modality was associated only with lymphopenia in the tumors of the lower esophagus, but not for those in the upper or middle esophagus. Increased target volume and radiation in the lower esophageal area increases the dose exposure of lymphocyte-harboring organs, such as the spleen, lung, liver, and stomach [63]. This is also supported by the findings of Davuluri et al. (2017) that the mean body dose exposure is the strongest factor associated with G4L development [72]. All of these findings suggest the potential of PBT in the reduction of lymphopenia in esophageal cancer patients undergoing RT.

## 5. Ongoing Clinical Trials

Randomized clinical trials are still lacking for PBT as a treatment for esophageal cancer. A randomized phase III study (NRG GI-006) by Lin et al. (2020) is currently accruing patients. Aside from the clinical outcome comparison between PBT and IMRT for patients with stage I–IVA esophageal cancer, NRG GI-006 will also assess the symptom burden, the effects on functional status, and quality of life [53]. It also aims to conduct a cost–benefit economic analysis comparing PBT and IMRT (NCT03801876) [53]. Recruiting patients to undergo PBT may however pose a challenge, due to the perceptions of IMRT superiority over PBT. In addition, insurance problems, as previously mentioned, may also arise.

Another on-going phase II study (NCT01684904) on the other hand is investigating the feasibility, toxicity, and efficacy of a regimen incorporating carboplatin/paclitaxel, with PBT, followed by definitive surgery. This investigator-designed regimen attempts to improve on the standard cisplatin/5FU regimen by adding a full-dose of paclitaxel and substituting carboplatin for cisplatin. Paclitaxel is a potent radiosensitizer with activity against advanced esophageal cancer, while carboplatin reduces the toxicity of the combination regimens, similar to that used by the CROSS trial. Added with the expected dose distribution patterns using PBT, this regimen is hypothesized to improve OS, safety, and tolerability of the treatment. Using the similar regimen design, a separate phase I dose escalation of neoadjuvant PBT with concurrent chemotherapy in locally advanced esophageal cancer (NCT02213497) is also ongoing, with the purpose of measuring the number of AEs within 3 years after treatment. Both trials, however, are only being performed in a small population of 38 and 30 participants, respectively.

## 6. Challenges and Limitations of PBT

### 6.1. PBT Planning

The dose distribution of the proton beam is highly dependent on the tissue material and density changes, much more than the photon beam [29]. Intra-fraction anatomical changes due to periodic tumor motion, diaphragmatic motion, and inter-fraction variation in patient anatomy due to set-up variability, weight loss, pleural effusions, stomach distension, or tumor changes, all pose a challenge to the treatment of esophageal cancer with PBT [29,76].

Several planning solutions may be implemented for a robust delivery of PBT, such as the use of four-dimensional computed tomography (4D-CT) simulation to characterize the respiratory motion of the target and normal tissues to produce a more robust and interplay-resistant plan with a comparable dose distribution for normal tissues [77,78], use of a 4D dataset to design the target volume, with the final plan evaluated on the end inspiratory and expiratory phases during free breathing-computed tomography (CT) techniques [79], use of maximum monitor-unit threshold-based iso-layered repainting technique, use of beam gating strategies [78], and use of abdominal compression to reduce diaphragmatic excursion [80].

Several mean dose predictive parameters could be used as guides in the planning of proton beam treatment. In a retrospective study comparing PS-PBT against IMRT in 55 patients with esophageal cancer, given 50.4 Gy in 28 fractions, Wang et al. (2015) was able to identify two geometric parameters which could be used to estimate the expected mean heart/lung doses and identify the dosimetric outliers, (1) distance from the planning target volume (PTV) to carina (DPC), and (2) the percentage of uninvolved heart (%UIH). The clinical target volume (CTV) was defined as “gross tumor volume (GTV; esophageal tumor plus involved nodes), plus elective coverage of the first echelon nodes and, occasionally, the celiac axis for distal tumors, plus a 3- to 4-cm superior/inferior margin, plus a 1.0- to 1.5-cm radial margin” [35]. The PTV in this study was generally a 0.5-cm expansion of the CTV. The DPC was found to be highly associated with the total lung mean dose for both IMRT (correlation value (cv): −0.83) and PBT (cv: −0.80). On the other hand, the %UIH was found to be significantly associated with the heart mean dose for both IMRT (correlation value (cv): −0.80) and PBT (cv: −0.71). This makes both DPC and %UIH great parameters to estimate the expected lung and heart mean doses, respectively, and presents utility in identifying the dosimetric outliers for both the PBT and IMRT plans [35].

The proton beam angles should also be chosen to spare high-risk organs. Zhang et al. (2008) previously utilized 4D-CT-based treatment planning to compare IMRT with 2-beam (AP/PA) and 3-beam (AP/2 posterior oblique beams) PBT for 15 esophageal cancer patients. There was no improvement in the cardiac sparing in either of the PBT plans compared to IMRT, despite an improvement in the median lung volume exposure [31]. The results from the study of Wang et al. (2015) suggest that the AP/PA beam approach may, in certain clinical situations, result in a suboptimal beam arrangement, with the AP component probably playing a larger role [35]. This is supported by a study completed by Zeng et al. (2016), wherein beam plans of PBS-PBT without an AP approach (PA plus left posterior oblique (LPO), and single PA) resulted in a lower stomach, liver, heart, and lung mean dose compared to the AP/PA beam [81]. However, in situations in which the PBT field encompasses the level of the diaphragm, a posterior-oblique field may be influenced by the respiratory motion of the diaphragm, such that the AP/PA beams may be more appropriate [78,82,83]; therefore, beam positioning should consider each patient’s specific clinical circumstances and institutional expertise. Other parameters that result in suboptimal PBT dosimetry include equal beam weighting (1:1), and unique anatomy (extension of CTV into or around the heart) [35]. Additionally, the use of angles between 180° and 220° must be considered [78]. These parameters can all be modified for re-optimization. When a unique anatomy is present, the pre-planning evaluation of the CTV margins is warranted with considerations of using the customized beam arrangements [35].

The evaluation for inter-fraction anatomic or setup changes during the treatment course should be completed, with periodic verification CT scans. Alternatively, daily volumetric imaging may be completed, using in-room CT-on-rails that will also be used during the actual image-guided RT. This can also guide the dose recalculation for an accurate Hounsfield unit [29,84]. This system, however, prolongs treatment time and potentially decreases the overall practice efficiency [84].

### 6.2. Issues of Equity

Global inequities in PBT cancer trials exist. Dodkins and colleagues found large mismatches between the global burden of disease and trials’ geographic location and target disease site [85,86]. For esophageal cancer, Eastern Asia exhibits the highest regional incidence rates, largely due to the burden in China [1]. The clinical trials, however, examining the use of PBT in esophageal cancer are based in Japan or the United States, with far fewer in the Chinese Clinical Trial Registry. The prospective trials include a clinical study of radiation-induced lung injury after proton and carbon-ion radiotherapy (ChiCTR-OOC-16009654) and a multi-center prospective clinical study for the strategy of surgery combined with proton heavy-ion radiotherapy and chemotherapy treatment of head and neck soft tissue sarcomas (ChiCTR-OIN-17011981). Most of the clinical trials are also executed in high-income countries (HIC), despite the increasing cancer burden in low- and middle-income countries (LMICs). This is partly because, out of the 116 functional proton/ion facilities in the International Atomic Energy Agency (IAEA) Directory of Radiotherapy Centers, 107 are located in HICs, eight (8) in upper-middle-income countries, and only one (1) in a LMIC. PBT is used in 98 of these centers [87,88]. India was the first LMIC and Southern Asian country to have a functioning proton cancer center established in January of 2019 [89]. In Southeast Asia, Thailand was the first and, so far, the only country yet that has opened a PBT center in February 2022 [90], in addition to a planned center in Singapore [91]. In addition, the trials from LMIC are less likely to be funded or published in high impact journals [86,92]. The insurance policies also contribute to this issue. It is expected that recruitment and continued participation in trials of the patients from LMIC would be more challenging, given the economic burden of PBT left to the participants to shoulder [53].

These inequities call for the need for international collaborative trials to assess these treatments, if we are to obtain more generalizable data [93]. In medicine, we generally assume that the result from one study is applicable to another group of individuals, without looking into the differences in context between the two. However, this may not always be true and just a “myth of generalizability” [94]. The results of Western clinical trials may not be totally applicable to patients in Asia. Future PBT studies should be multinational in scope if we really want to extend the global reach of this therapy.

## 7. Conclusions

PBT has demonstrated dosimetric advantages compared with photon-based techniques. The limited studies on the translation of these results to clinical outcome supports that PBT leads to improved disease-related outcomes and reduction in treatment-related AEs. More large-scale trials must be completed in the future to contribute to the growing literature of PBT. Newer results from ongoing clinical trials will dictate the future utility of PBT versus photon-based RT. However, with the current results, PBT offers an opportunity for the re-evaluation of treatment intensification in cancer treatment and toxicity mitigation, incorporating the role of immunotherapy. The global collaboration of investigators and industries for PBT clinical trials is encouraged, to generate results that are more equitable and globally generalizable.

## Figures and Tables

**Table 1 cancers-14-04045-t001:** Dosimetric studies comparing photon-based and proton beam therapy techniques for esophageal cancer treatment. This table builds upon studies reviewed in Jethwa et al. (2020) [29].

Author (Year)	Country/Countries of Origin	Number of Patients	Race/Ethnicity Breakdown (n)	Cancer Features Breakdown (n)	Comparison Arms	Dose (Gy)/Number of Fractions	Dosimetric Advantages of PBT
Zhang et al. (2008) [31]	United States	15	-	Location: distal/GEJ; Stage not stated	PS-PBT vs. IMRT	50.4/28	Heart: V40–50 Gy. Lung: Dmean, V5–20 Gy
Welsh et al. (2011) [34]	United States	10	-	Location: distal; Stage not stated	PBS-PBT vs. IMRT	65.8/28	Lung: Dmean, V5–20 Gy
Wang et al. (2015) [35]	United States	55	-	Location: upper (2), mid (11), distal (42); Stage not stated	PS-PBT vs. IMRT	50.4/28	Heart: Dmean, V10–30 Gy Lung: Dmean, V5–20 Gy
Warren et al. (2016) [36]	United Kingdom	21	-	Location: midthoracic	PBS-PBT vs. VMAT	50–62.5/25	Heart: Dmean, V5 Gy Lung: Dmean
Shiraishi et al. (2017) [33]	United States	727	-	Location: mid (76), distal (651); Stage I (18), IIA (225), IIB (29), III (423), IVA (32)	PS-PBT (99%) or PBS-PBT vs. IMRT	50.4/28	Heart: Dmean, V5–40 Gy Lung: Dmean, V5 Gy Liver: Dmean, V30 Gy
Xi et al. (2017) [32]	United States	343	Ethnicity: Caucasian (290), others (53)	Location: upper/mid (95), distal/GEJ (248); Stage I/II (117), III (226)	PS-PBT (95%) or PBS-PBT vs. IMRT	50.4/28	Heart: Dmean, V30 Gy Lung: Dmean, V5–20 Gy
Hirano et al. (2018) [37]	Japan	27	-	Location: upper (5), mid (9), mid/distal (6), distal (5), distal/abdominal (2); Stage IIIA (15), IIIB (9), IIIC (3)	PBS-PBT vs. 3DCRT or IMRT	60/30	Heart: Dmean, V10–40 Lung: Dmean, V5–20 Gy
Macomber et al. (2018) [38]	United States	55	-	Location: mid (10), distal/GEJ (45); Stage IIA (3), IIB (16), IIIA (30), IIIB (5), IIIC (1)	Mixed PS-PBT and PBS-PBT vs. IMRT vs. 3DCRT	50.4/28	Heart: Dmean, V5–40 Gy
Liu et al. (2019) [39]	United States	35	-	Location: distal; Stage not stated	PBS-PBT vs. VMAT	50–50.4/25–28	Heart: Dmean, V20 Gy Lung: Dmean, V5 Gy Liver: Dmean, V30 Gy

GEJ, gastroesophageal junction; PBT, proton beam therapy; IMRT, intensity modulated radiotherapy; PS-PBT, passively scattered proton beam therapy; PBS-PBT, pencil beam scanning proton beam therapy; VMAT, volumetrically modulated arc therapy; 3DCRT, 3-dimensional conformal radiotherapy; Dmean, mean dose to entire organ; Vx Gy, percentage volume of organ receiving at least X Gy.

**Table 2 cancers-14-04045-t002:** Clinical studies comparing PBT vs. IMRT for esophageal cancer treatment. This table builds upon studies reviewed in Jethwa et al. (2020) [29].

Author (Year)	Country/ Countries of Origin	Study Design	Number of Patients	Race/Ethnicity Breakdown (n)	Cancer Features Breakdown (n)	Technique/ Modality	Dose (Gy)/Number of Fractions	Outcomes of PBT (IMRT)	Toxicity of PBT (IMRT)	Post-Op Complications of PBT (IMRT)
Lin et al. (2012) [51]	United States	Retrospective cohort	62	Ethnicity: Caucasian (59), African American (1), Hispanic (1), Asian (1)	Location: upper (3), mid (11), distal/GEJ (48); Stage I (2), II (20), III (32), IVA (2), IVB (6)	PS-PBT	50.4/28	OS3: 52%; RFS3: 41%; DMFS3: 67%; LRC3: 57%	G2+ pneumonitis: 3%; mortality: 3%	Pulmonary: 7%; cardiac: 8%; anastomotic leak: 7%; wound: 3%
Ishikawa et al. (2015) [50]	Japan	Retrospective cohort	40	-	Location: cervical (2), upper (10), mid (21), distal (7); Stage I (16), II (9), III (15)	PS-PBT	60/30	OS2: 75%; LRC2: 66%; CSS2: 77%	G3+ pulmonary: 0%; G3+ cardiac: 0%	–
Xi et al. (2017) [32]	United States	Retrospective cohort	343	Ethnicity: Caucasian (290), others (53)	Location: upper/mid (95), distal/GEJ (248); Stage I/II (117), III (226)	PBS-PBT/PS-PBT; IMRT	50.4/28	OS5: 42% (32%) *; PFS5: 35% (20%) *; DMFS5: 65% (50%) *	G3+: 38% (45%)	–
Lin et al. (2017) [52]	United States	Multi-institutional retrospective cohort	580	-	Location: upper/mid (41), distal/GEJ/cardia (539); Stage I/II (211), III/IV (369)	PBS-PBT/PS-PBT; IMRT; 3DCRT	50.4/28	–	–	Pulmonary: 16% (24%) *; cardiac: 12% (12%); wound: 5% (14%) *; GI: 19% (23%); hospital stay: 9 [12] days *
Hirano et al. (2018) [37]	Japan	Retrospective cohort	27	-	Location: upper (5), mid (9), mid/distal (6), distal (5), distal/abdominal (2); Stage IIIA (15), IIIB (9), IIIC (3)	PBS-PBT	60/30	OS1: 90.8% PFS1: 40.6%	G3+ Esophagitis: 4% G2+ cardiac: 19%	-
Ono et al. (2019) [48]	Japan	Multicenter retrospective cohort	202	-	Location: cervical (20), thoracic (181), abdominal (1); Stage I (72), II (30), III (52), IV (48)	PBT	BED10 87.2	OS5: 56.3% LRC: 64.4%	G3+ cardiac: 1% G3+ pulmonary: 0.5%	-
Lin et al. (2020) [53]	United States	Prospective randomized phase IIb trial	107	Race: White (98), Black (2), Asian (1), unknown (6); Ethnicity: Not Hispanic (95), Hispanic (11), unknown (1)	Location: upper (3), mid (15), distal (98); Stage I (6), II (41), III (60)	PS-PBT/PBS-PBT; IMRT	50.4/28	PFS3: 51.2% (50.8%) OS3: 44.5% (44.5%)	TTB: 17.4 (39.9)	POCS: 2.5 (19.1)

*, indicates statistically significant improvements with PBT compared to photon-based technique. GEJ, gastroesophageal junction; PBT, proton beam therapy; IMRT, intensity modulated radiotherapy; PS-PBT, passively scattered proton beam therapy; PBS-PBT, pencil beam scanning proton beam therapy; 3DCRT, 3-dimensional conformal radiotherapy; GI, gastrointestinal; OS, overall survival; LRC, locoregional control; CSS, cancer-specific survival; G, grade; PFS, progression-free survival; DMFS, distant metastasis free survival; TTB, total toxicity burden; POCS, post-operative complication severity score; BED, biological effective dose.

**Table 3 cancers-14-04045-t003:** Selected studies comparing development of treatment related lymphopenia (TRL) with photon-based therapy vs. PBT for esophageal cancer treatment.

Author (Year)	Country/Countries of Origin	Number of Patients	Treatment Arm	Dose (Gy)/Number of Fractions	Lymphopenia
Fang et al. (2017) [63]	United States	220	IMRT vs. PBT	45–50.4/25–28	47.27% vs. 30.9% (AOR: 2.13)
Davuluri et al. (2017) [72]	United States	504	IMRT vs. PBT	50.4/28	33% vs. 16%
Shiraishi et al. (2018) [62]	United States	480	IMRT vs. PBT	50.4/28	40.4% vs. 17.6% (OR: 3.45)
Routman et al. (2019) [64]	United States	144	3DCRT or IMRT vs. PBT	41.4–50.4/23–28	56% vs. 22% (OR: 5.13)
Zhou et al. (2019) [75]	China	286	CRT	50–60/28–30	31%

Lymphopenia in general is associated with worse clinical outcomes in cancer patients [61,71,72]. PBT, proton beam therapy; IMRT, intensity modulated radiotherapy; 3DCRT, 3-dimensional conformal radiotherapy; TRL, treatment-related lymphopenia; OR, odds ratio; AOR, adjusted odds ratio.

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
