# Peer review of "Proton Beam Therapy for Esophageal Cancer"

_cancers, 2022, doi:10.3390/cancers14164045_

Round 1
Reviewer 1 Report
The PBT for esophageal cancer is important issue. However, there are some points to revise before publishing.
1. There are some reviews about PBT for esophageal cancer. What the unique point of your review is?
2. Authors wrote “chemoradiotherapy and 61 salvage surgery is a logical first-line treatment”. I think so too, but there are several treatment schemes. So, this sentence is exaggeration.
3. As authors said in line 90, IMRT needs longer time than 3DCRT, but PBT also needs longer time than 3DCRT. So, treatment time is not important problem for choosing PBT. What do you think about it?
4. I do not understand what the role of paragraph of lines 138-166. It is advantage of PBT for another caner.
5. You said that “it is still necessary to irradiate a wider field even if no lymph node metastases are radio-175 logically detected prior to treatment”. However, whether elective nodal irradiation is needed or not is very controversy. What do you think about it?
6. I agree with the importance of lymphocytopenia for prognosis. So, could you make new table about it for reader?
You wrote cGY in line 355. Do you mean Gy?
Reviewer 2 Report
This article comprehensively reviewed proton beam therapy for localized esophageal cancer with aspect of dosimetric comparison with photon radiotherapy, toxicities, treatment-related lymphopenia, as well as on-going clinical trials. The focus is sufficiently novel and interesting.
Major
In 6.1 PBT planning section, the combination of PA and posterior-oblique beam looks better to spare heart than AP/PA parallel opposite beams.
But, when proton beam field encompass the level of diaphragm, a posterior-oblique beam can be influenced by respiratory motion of diaphragm. Therefore, we prefer AP/PA beam, if proton beam field include the level of diaphragm.
What do you think?
Minor
1. Please check the sentence on lines 88-89.
2. On line 355, check 50.4 cGY.
